# Somatic Copy Number Alterations in Colorectal Cancer Lead to a Differentially Expressed ceRNA Network (ceRNet)

**Héctor Herrera-Orozco** [1,2], **Verónica García-Castillo** [1], **Eduardo López-Urrutia** [1],
**Antonio Daniel Martinez-Gutierrez** [3], **Eloy Pérez-Yepez** [3], **Oliver Millán-Catalán** [3], **David Cantú de León** [3],
**César López-Camarillo** [4], **Nadia J. Jacobo-Herrera** [5], **Mauricio Rodríguez-Dorantes** [6],
**Rosalío Ramos-Payán** [7] **and Carlos Pérez-Plasencia** [1,3,*]

1    Laboratorio de Genómica, FES-Iztacala, Universidad Nacional Autónoma de México. Av. De los Barrios 1, Los
     Reyes Iztacala, Tlalnepantla 54090, Mexico; hhgamer@gmail.com (H.H.-O.); garciaver@gmail.com (V.G.-C.);
     e_urrutia@unam.mx (E.L.-U.)
2    Posgrado en Ciencias Biológicas, Universidad Nacional Autónoma de México, Edificio D. Circuito de
     Posgrados, Ciudad Universitaria, Coyoacán, Mexico City 04510, Mexico
3    Laboratorio de Genómica, Instituto Nacional de Cancerología, Av. San Fernando 22, Tlalpan,
     Mexico City 14080, Mexico; maga94@comunidad.unam.mx (A.D.M.-G.); eperezy2306@gmail.com (E.P.-Y.);
     oliver.millan.sg@gmail.com (O.M.-C.); dfcantu@gmail.com (D.C.d.L.)
4    Posgrado en Ciencias Genómicas, Universidad Autónoma de la Ciudad de México, Calle Dr. García Diego
     168, Cuauhtémoc, Mexico City 06720, Mexico; cesar.lopez@uacm.edu.mx
5    Unidad de Bioquímica, Instituto Nacional de Ciencias Médicas y Nutrición Salvador Zubirán, Av. Vasco de
     Quiroga 15, Tlalpan, Mexico City 14080, Mexico; nadia.jacobo@gmail.com
6    Laboratorio de Oncogenómica, Instituto Nacional de Medicina Genómica, Tlalpan,
     Mexico City 14610, Mexico; mrodriguez@inmegen.gob.mx
7    Faculty of Chemical and Biological Sciences, Autonomous University of Sinaloa, Culiacan 80030, Mexico;
     rosaliorp@uas.edu.mx
*    Correspondence: carlos.pplas@gmail.com or carlos.pplas@unam.mx; Tel.: +52-(55)-56231333 (ext. 39807)

**Abstract:** Colorectal cancer (CRC) represents the second deadliest malignancy worldwide. Around 75% of CRC patients exhibit high levels of chromosome instability that result in the accumulation of somatic copy number alterations. These alterations are associated with the amplification of oncogenes and deletion of tumor-ppressor genes and contribute to the tumoral phenotype in different malignancies. Even though this relationship is well known, much remains to be investigated regarding the effect of said alterations in long non-coding RNAs (lncRNAs) and, in turn, the impact these alterations have on the tumor phenotype. The present study aimed to evaluate the role of differentially expressed lncRNAs coded in regions with copy number alterations in colorectal cancer patient samples. We downloaded RNA-seq files of the Colorectal Adenocarcinoma Project from the The Cancer Genome Atlas (TCGA) repository (285 sequenced tumor tissues and 41 non-tumor tissues), evaluated differential expression, and mapped them over genome sequencing data with regions presenting copy number alterations. We obtained 78 differentially expressed (LFC > 1 | < −1, padj < 0.05) lncRNAs, 410 miRNAs, and 5028 mRNAs and constructed a competing endogenous RNA (ceRNA) network, predicting significant lncRNA–miRNA–mRNA interactions. Said network consisted of 30 lncRNAs, 19 miRNAs, and 77 mRNAs. To understand the role that our ceRNA network played, we performed KEGG and GO analysis and found several oncogenic and anti-oncogenic processes enriched by the molecular players in our network. Finally, to evaluate the clinical relevance of the lncRNA expression, we performed survival analysis and found that C5orf64, HOTAIR, and RRN3P3 correlated with overall patient survival. Our results showed that lncRNAs coded in regions affected by SCNAs form a complex gene regulatory network in CCR.

**Keywords:** colorectal cancer; somatic copy number alterations; long non-coding RNAs; competitive endogenous RNA (ceRNA) network

## 1. Introduction

Colorectal cancer (CRC) represents the third most common and the second deadliest malignancy worldwide. In 2020, there were more than 1.9 million new colorectal cancer cases and more than 900,000 CRC-associated deaths worldwide [1]. The number of CRC cases is currently rising and is estimated to reach 2.5 million by the year 2035 [2]. Around 75% of CRC cases are associated with high levels of chromosome instability (CIN), which results in aneuploidies, amplifications or deletions, and even somatic copy number alterations (SCNAs) [3]. CIN leads to tumor growth by increasing the phenotypic variation upon which natural selection can act, eliciting the acquisition of adaptations that drive tumor evolution [4].

SCNAs are sub-chromosomal somatic alterations that result in oncogene or tumor-suppressor copy gains or losses in different types of cancer [5]. SCNAs that confer adaptations occur throughout tumor development from the early stages [6] to favor transformation from in situ carcinoma to invasive tumors [7] and even metastases development [8].

SCNAs affect not only the coding genome but the non-coding transcriptome as well [9]. One of the most important groups of non-coding RNAs (ncRNAs) is long ncRNAs (lncRNAs), transcripts longer than 200 nts that generally lack an open reading frame (ORF) and thus cannot be translated into proteins [10]. lncRNAs can interact with multiple targets, acting as regulators of gene expression along the flow of genetic information [11].

Alterations that target lncRNAs can strongly impact the development of multiple types of cancer. One of the main gene-regulating roles of lncRNAs is acting as miR-sponges, sequestering multiple miRNAs, preventing their binding to mRNAs, and downregulating mRNA translation [12]. For example, CDC6 promotes breast cancer progression by sponging miR-215 [13]; another example is DANCR, which promotes lung cancer by sequestering miR-216a [14]. Several lncRNAs are even potential CRC progression biomarkers, as recent research shows [15].

Both mRNAs and lncRNAs harbor multiple miRNA-binding sequences, thus competing for miRNA binding in what are called competing endogenous RNA networks, or cerR-Nets [16]. These networks are associated with different cancer types, such as esophageal cancer [17], lung adenocarcinoma [18], and hepatocellular carcinoma [19]. Chromosomal alterations can affect ceRNnets, as reported in breast and lung cancer [20,21].

There are several reports of ceRNA networks in CRC that control processes such as tumor phenotype promotion [22], proliferation [23], and autophagy [24]. These ceRNets are even associated with patient staging [25] and proposed as potential biomarkers [26]. Nevertheless, the effect that SCNAs have in a lncRNA-mediated ceRNA network in CRC has not been described. Therefore, in this work, we identified differentially expressed lncRNAs coded in genomic regions affected by SCNAs in CRC and constructed a ceRNA network to evaluate their possible involvement in CRC. Our ceRNA network comprised 33 differentially expressed lncRNAs and 76 mRNAs partaking in several KEGG pathways from which proteoglycans in cancer and bladder cancer stood out. Through Kaplan–Meier analyses, we found that the lncRNAs HOTAIR and RRN3P3 are clinically relevant for CRC patients due to their association with lower overall survival and a poorer prognosis, respectively. These results showed the participation of SCNA-associated lncRNAs in the CRC phenotype, strengthening the importance of ncRNA research.

## 2. Materials and Methods

To identify the differentially expressed long non-coding RNAs (DElncRNAs) coded in genomic regions with somatic copy number alterations (SCNAs) that arose during tumor progression in the colorectal cancer (CRC) patient samples and the effect they had on gene expression regulation, we proposed a bioinformatics workflow that used several data mining methods associated with the R programing (4.1.1) language and the Bioconductor (3.13) environment [27]. This workflow is outlined in Figure 1.

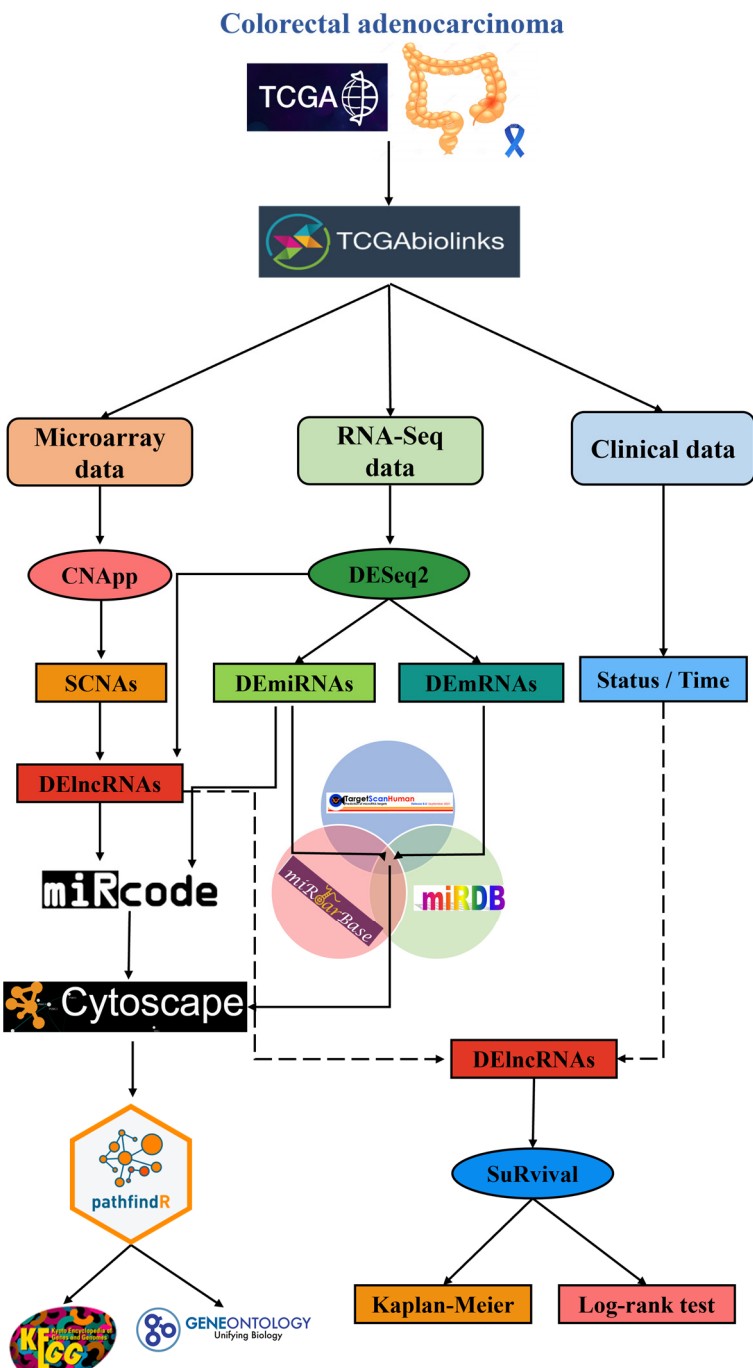

**Figure 1.** Graphical summary of the bioinformatic workflow. Microarray, RNA-Seq, and clinical data were sourced from TCGA, processed, normalized, analyzed, and then cross-referenced to obtain the proposed ceRNA network. Finally, pathway and survival analyses for said network were performed.

## 2.1. RNA-Seq

The first goal was to acquire the list of DElncRNAs. We used the TCGAbiolinks package in R [28], which allowed us to access The Cancer Genome Atlas (TCGA) repository [29] and download the RNA-seq expression files from the 326 patients (285 primary tumors, 41 healthy samples) available for the Colorectal Adenocarcinoma Project.

## 2.2. BiomaRt

The next goal was to determine which of the transcripts were lncRNAs. We used the BiomaRt package [30] to obtain the external gene name and the biotype of the genes. We

selected all biotypes that fulfilled the lncRNA description (an RNA transcript with a length of more than 200 nt and no coding potential), which were the following: "lncRNA", "polymorphic pseudogene", "transcribed processed pseudogene", "transcribed unprocessed pseudogene", "unprocessed pseudogene", "processed pseudogene", "transcribed unitary pseudogene", and "unitary pseudogene".

### 2.3. Differential Expression

Once we had the expression matrix with the lncRNAs, the next step was finding which transcripts were differentially expressed in the tumor samples. To achieve this, we employed the DESeq package [31] and fed the expression matrix and the sample metadata (i.e., whether samples were healthy or tumoral) into this package. We used log2 fold change (LFC) > 1 as an indicator of positive alteration and LFC < −1 as an indicator of negative alteration. In both cases, we filtered lncRNAs by padj < 0.05 (two-sided) for statistical significance.

### 2.4. Hierarchical Clustering

Next, we wanted to study the patterns of differentially expressed genes between the tumor and healthy samples. For this purpose, we took the raw-count lncRNA matrix and used the variance stabilizing transformation from the DESeq package to normalize our dataset. Afterward, we calculated the Z score per gene and used the Complex Heatmap package [32] to draw the expression heatmaps.

### 2.5. Somatic Copy Number Alterations

Once we distinguished which lncRNAs were differentially expressed, the next step was to determine which ones modified their expression because they were coded in a genomic region with somatic copy number alterations. To achieve this, using the TCGAbiolinks package, we downloaded the genomic sequencing data of the Colorectal Adenocarcinoma Project. The files contained sequencing experiments from 545 patients (536 primary tumors, 8 healthy samples). The sequencing files were uploaded to CNApp [33], a free web-based software coded in R, which re-segmented the samples and returned a map of the regions that were amplified or deleted in the dataset, indicating the frequency of each alteration. This platform also returned a list of genes coded in each region. The criteria for selecting the DElncRNAs were presenting an LFC > 1 and being coded in a region that was altered in at least 10% of the samples [34].

### 2.6. DElncRNA–DEmiRNA Interactions

To understand the role of the DElncRNAs in regulating gene expression, we generated a ceRNet. The first step towards constructing the network was to identify the DElncRNA–DEmiRNA interactions from the interactions predicted in the MiRCode database [35].

### 2.7. DEmiRNA–DEmRNA Interactions

The next step was determining which mRNAs were targeted by our DEmiRNAs to find interactions with a greater impact on CRC progression. We used three different databases to filter them: (1) MiRTarBase [36]; (2) MiRDB [37]; and (3) TargetScan [38].

We downloaded and searched for the DEmiRNA–DEmRNA interactions in each of the three databases. Next, we contrasted the shared interactions among all three databases using the VennDiagram package [39], which allowed us to find the overlap in the databases and thus generate DElncRNA–DEmiRNA–DEmRNA interactions.

### 2.8. Statistical Significance

To validate the DElncRNA–DEmiRNA–DEmRNA interactions, we assessed whether the lncRNA expression level was correlated with the mRNA expression level, following the rationale that a given lncRNA sequesters a miRNA that downregulates an mRNA, releasing the mRNA from the negative regulation, effectively upregulating it. For the correlations,

we used the Hmisc package [40], which allowed us to perform, in a single step, Pearson's correlations among all the possible interactions in the matrix.

Once we obtained the matrix with all correlations, we filtered only those that contained our DElncRNAs and DEmRNAs. We considered a *p* value < 0.05 as statistically significant and, since we assumed that our DElncRNAs downregulated the miRNAs to upregulate the DEmRNA expression, we only considered positive correlations for subsequent analysis (R > 0).

### 2.9. ceRNA Network

Once we had all positive and statistically significant interactions, we uploaded the resulting interaction matrix to Cytoscape v.3.9.1 software [41] to visualize the network.

### 2.10. Gene Ontology and KEGG

To analyze the effect that the network had on the CRC phenotype, we evaluated the enriched pathways and biological processes. To perform both the KEGG and GO analyses, we employed the R package pathfindR [42] using a list of the gene names of the mRNAs present in the ceRNA network, their LFCs, and the padj obtained from the DESeq2 package.

### 2.11. Survival Curves

To assess the relationship between the lncRNA expression and the clinical outcome of the patients, we performed a survival analysis with the Survival package in R [43]. First, we divided the expression of each ceRNet-associated lncRNA by the median and ran Kaplan–Meier analyses for each half of the sample. We assessed statistical significance using a log-rank and cumulative hazard test with Cox regression, and all patient data were followed-up at five and eleven years.

## 3. Results

We investigated the role of lncRNAs coded in genomic regions affected by somatic copy number alterations (SCNsA) in gene expression regulation through a bioinformatic approach using different tools from R software (4.1.1) and the Bioconductor environment.

We first distinguished which lncRNAs were expressed in CRC samples from the TCGA repository. Using the BiomaRt package and Deseq2 we obtained 135 upregulated lncRNAs and 133 downregulated lncRNAs, for a total of 268 DElncRNAs (LFC > 1 for a positive change, LFC < −1 for a negative change, and padj < 0.05 in both cases for statistical significance; Figure 2).

To identify which SCNA-associated lncRNAs altered their expression, we used genomic sequencing uploaded to the CNApp platform. We found several amplifications and deletions with a frequency in >10% of patients (Figure 3) and multiple alterations that affected >50% of patients. We identified 78 lncRNAs coded in a region altered in at least 10% of patients and with a corresponding LFC < −1 or LFC >1 (Table S1).

We evaluated the role of the SCNA-associated DElncRNAs in regulating genetic expression in CRC by constructing a competing endogenous RNA network (ceRNet, Figure 4, Table S2). First, we identified 410 DEmiRNAs, from which 187 were upregulated (LFC > 1, padj < 0.05) and 223 downregulated (LFC < −1, padj < 0.05; Figure S1). We repeated the procedure for mRNAs and found 5028 DEmRNAs, of which 2294 were upregulated and 2734 downregulated (Figure S2). Using different databases, we predicted interactions between DElncRNAs and DEmiRNAs and between DEmiRNAs and DEmRNAs, constructing a ceRNet with 30 DElncRNAs, 19 DEmiRNAs, and 77 DEmRNAs (Figure 4).

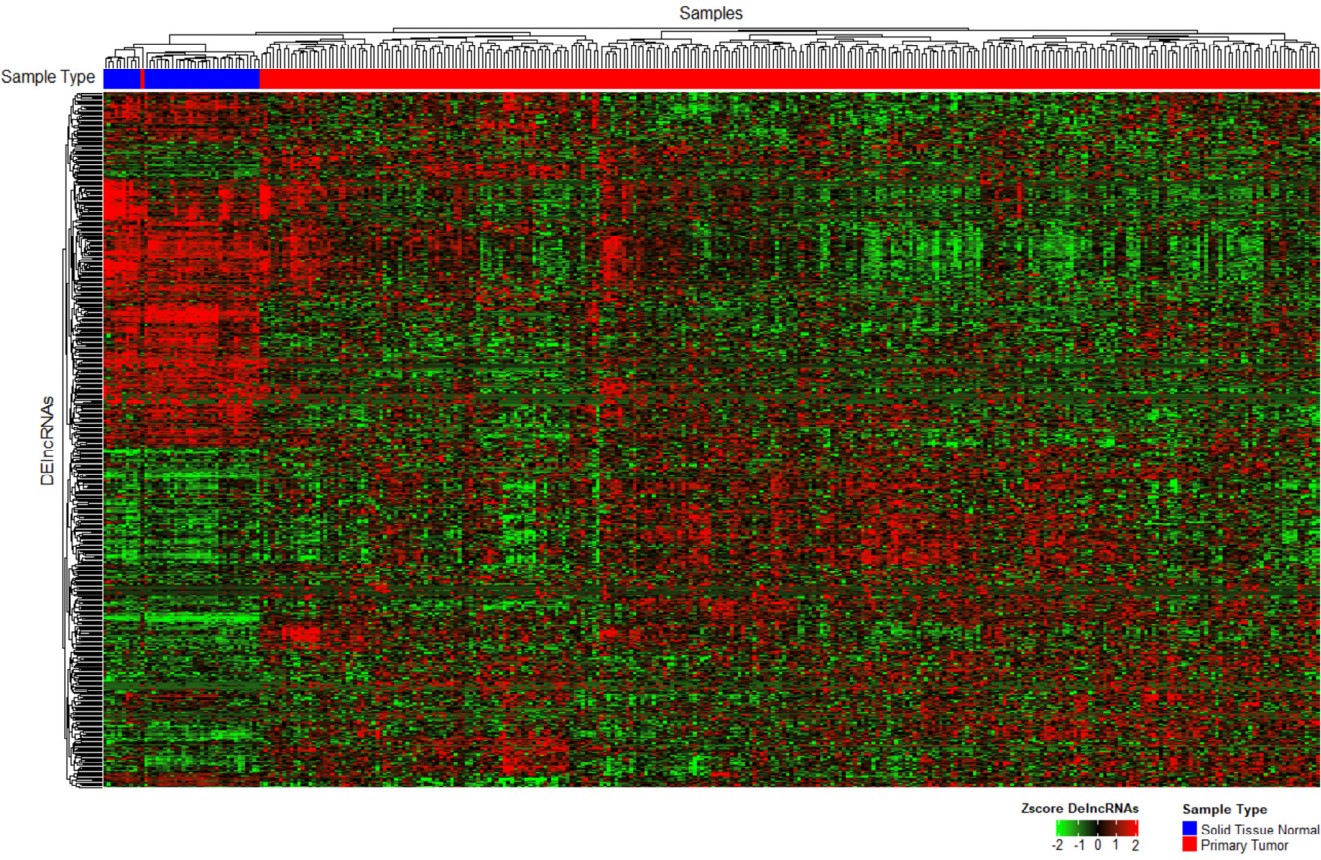

**Figure 2.** Unsupervised heatmap of the differentially expressed lncRNAs between healthy and tumor samples. The **top** row presents samples grouped by k-means clustering (distributed by color blue = healthy, red = tumor; padj < 0.05).

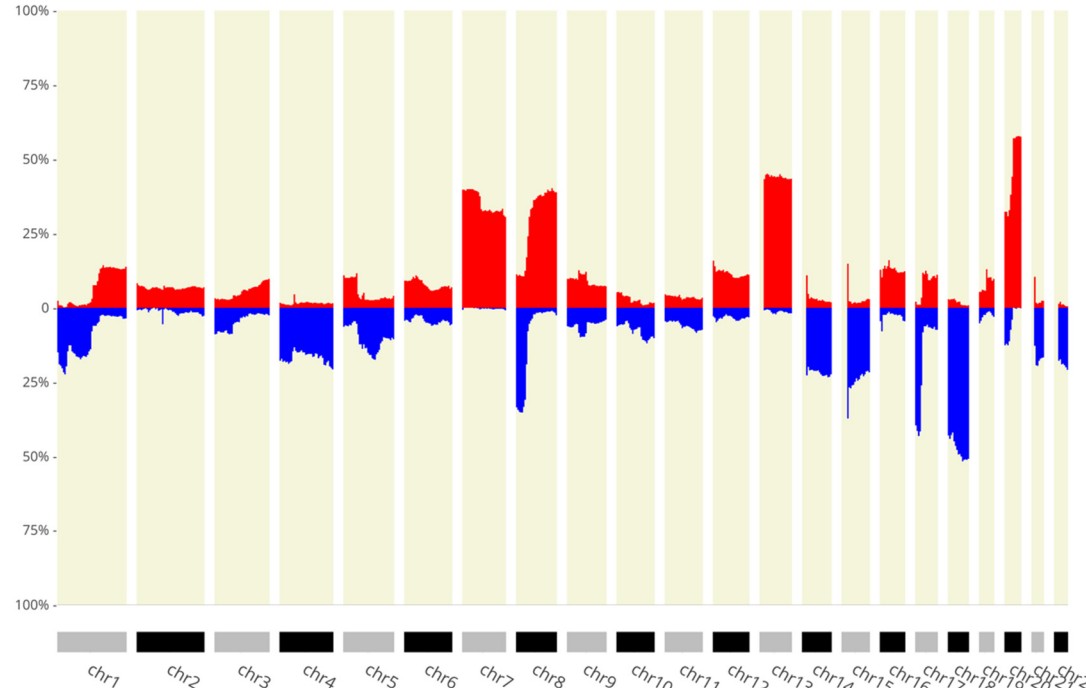

**Figure 3.** The x-axis shows the chromosomes, and the *y*-axis depicts the percentage of patients with the amplification/deletion feature. In blue we can see deletions and in red amplifications (*p* < 0.05).

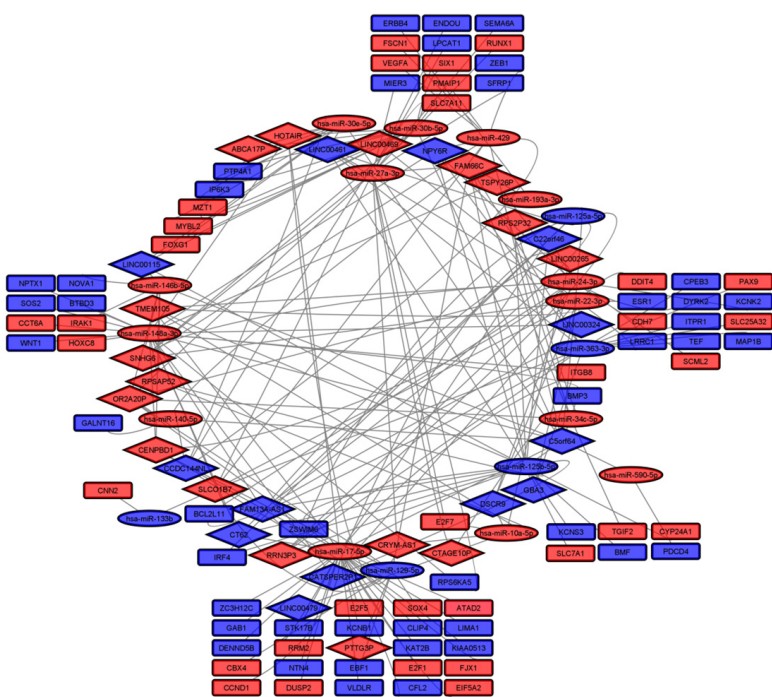

**Figure 4.** ceRNA network (blue = downregulated; red = upregulated; diamonds = lncRNAs; ellipses = miRNAs; rectangles = mRNAs).

To determine the involvement of the proposed ceRNA network in gene expression regulation, we performed a KEGG pathways analysis. The pathway with the most upregulated genes—10 genes—was proteoglycans in cancer; meanwhile, the pathways associated with bladder cancer were most enriched—almost 15-fold—when analyzing upregulated mRNAs (Figure 5A). Interestingly, the proteoglycans in the cancer pathway also had the most downregulated genes (eight genes), while renal cell carcinoma-associated pathways were the most enriched (more than 12-fold) when analyzing downregulated genes (Figure 5B). When analyzing up- and downregulated mRNAs together, proteoglycans in cancer remained the pathway with the highest number of DEmRNAs, while bladder cancer was the most enriched (Figure 5C).

The genes associated with the enriched pathways interacted with each other. The proteoglycans in the cancer pathway were closely connected with the calcium signaling pathway and the chemical carcinogenesis pathway and indirectly connected with the RAS signaling pathway and even the ERbB4 signaling pathways, among other important interactions shown in Figure 5D.

Regarding the GO analysis, we found upregulated mRNAs in multiple processes, mainly DNA-binding transcription activator activity (seven genes) and RNA polymerase II cis-regulatory region sequence-specific DNA (five genes). Two processes were enriched >30-fold: DNA-binding transcription activator activity and the PcG protein complex (Figure 6A). The GO analysis of downregulated mRNAs showed that cell migration (three genes), the mitochondrial outer membrane, delayed rectifier potassium channel activity, and cellular response to estradiol stimulus (two genes) had the most downregulated genes. Similarly, delayed rectifier potassium channel activity and cellular response to estradiol stimulus were enriched >50-fold (Figure 6B). The GO analysis using both up- and downregulated mRNAs together did not resemble either. The positive regulation of the apoptotic process had the most DEmRNAs (six genes), followed by cell migration and the positive regulation of the release of cytochrome c from mitochondria—both with five genes; in addition, this last process was the most enriched pathway (Figure 6C).

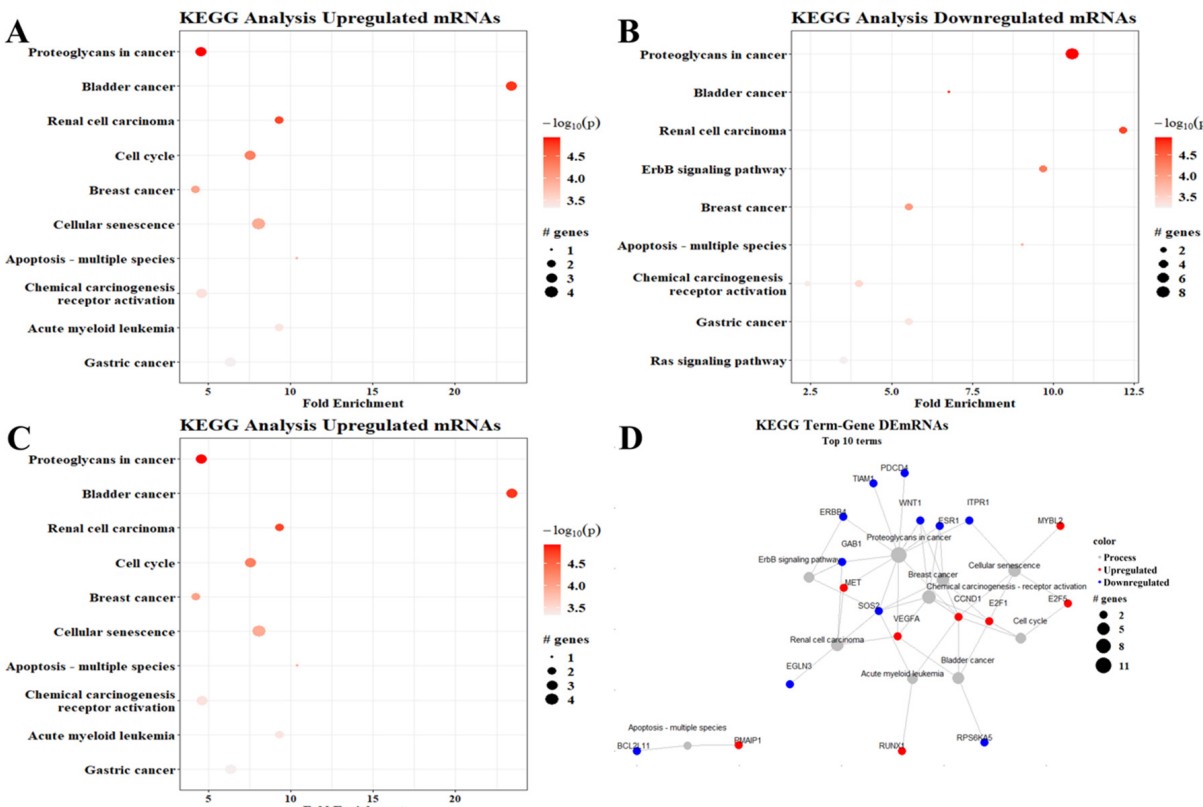

**Figure 5.** KEGG pathway enrichment analysis. (**A**) Upregulated mRNAs, (**B**) downregulated mRNAs, (**C**) all DEmRNAs, (**D**) term–gene graph. Number of genes altered per pathway and their fold enrichment; the graphs show the top ten processes observed in the analysis.

The term–gene interaction from the GO analysis showed a particularly close relationship between two processes: DNA-binding transcription activator activity and transcription factor binding. On the other hand, three processes were enriched but not interconnected: the PcG protein complex, protein tyrosine kinase activity, and stress fibers (Figure 6D). Furthermore, we could see that two of the most enriched processes were involved in the activity of transcription factors such as MYCN, MYBL2, and SOX2, as well as cell cycle regulators such as E2F1/5.

Finally, we estimated the possible clinical relevance of our DElncRNAs through Kaplan–Meier analyses, where overall survival was associated with the expression of the lncRNAs divided by the median (Figures 7 and S3–S6) with a follow-up at five and eleven years. We obtained two lncRNAs presenting a significant association with patient survival in both the 5-year and 11-year intervals: high HOTAIR (Figure 7A) expression was associated with lower overall survival (5 years $p = 0.0052$; 11 years $p = 0.039$) and a higher HR (5 years $p = 0.006$, 11 years $p = 0.0041$) and a lower RRN3P3 (Figure 7B) were associated with a poorer prognosis for the patients (5 years $p = 0.019$; 11 years $p = 0.053$) and a higher HR (5 years = 0.021; 11 years = 0.056). We also found three lncRNAs associated with patient prognosis at five years—CCDC144NL, LINC00479, and TSPY26P (Figures S3–S5)—and one lncRNA at the eleven-year mark—C5orf64 (Figure S5).

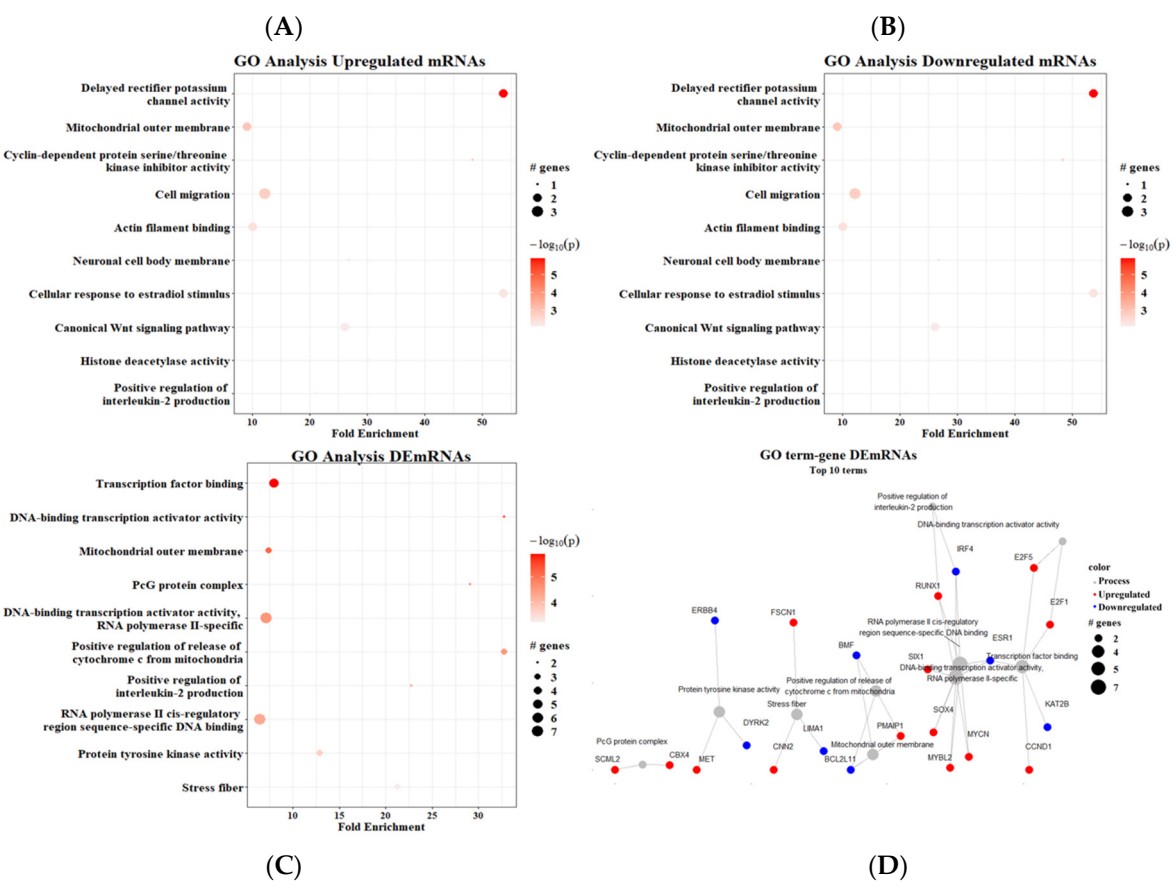

**Figure 6.** Gene ontology analysis. (**A**) Upregulated mRNAs, (**B**) downregulated mRNAs, (**C**) all DEmRNAs, (**D**) term–gene graph. Number of genes altered per pathway and their respective fold enrichment.

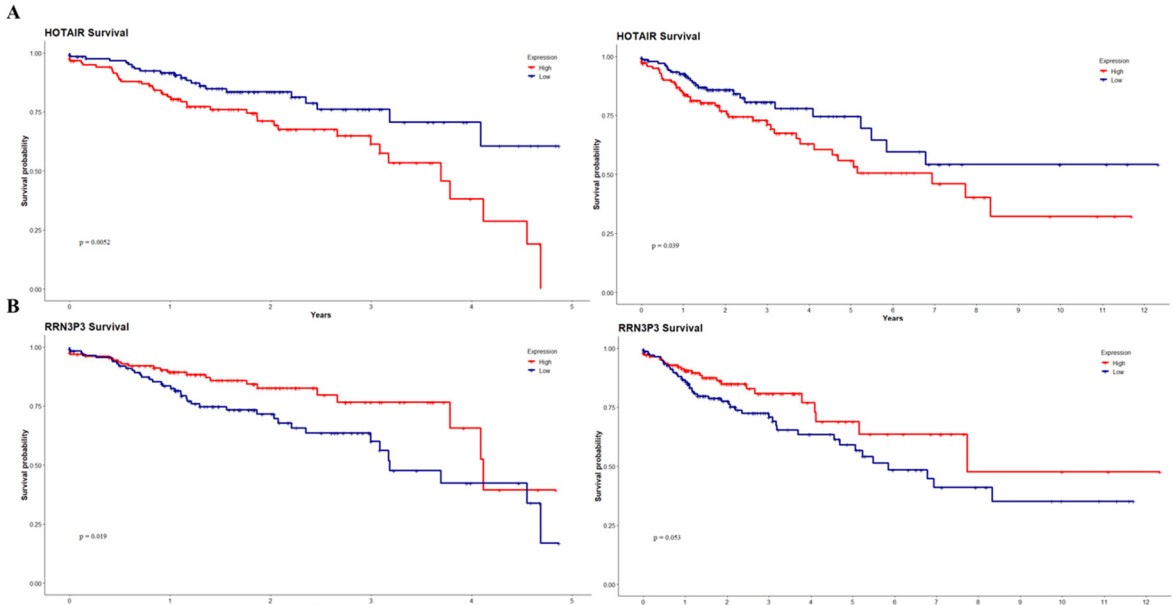

**Figure 7.** DElncRNA-associated survival curves up to 5 and 11 years. High HOTAIR expression (**A**) was associated with lower overall survival ($p = 0.0052$, $p = 0.039$) and a higher hazard ratio ($p = 0.006$, $p = 0.0041$). Lower RRN3P3 expression (**B**) was associated with lower overall survival ($p = 0.019$, $p = 0.053$) and a higher HR ($p = 0.021$; $p = 0.056$).

## 4. Discussion

With less than 5% of the human genome coding for proteins and another ~85% actively transcribed into different types of RNA [44], a myriad of questions arises regarding the complex gene regulation of cancer development. In the last 10 years, our understanding of lncRNA biology and functions has deepened and broadened, up to the point of recognizing lncRNAs as central players in cancer phenotype, progression, and even metastasis. Our findings (Figure 2) correspond to what has previously been reported: that tumorigenesis and cancer progression alter lncRNA expression, which, in turn, participates and regulates said processes [45]. Additionally, the magnitude of the differences in expression patterns between tumor and healthy samples was ample enough to show up in the unsupervised clustering algorithms.

This is supported by the growing body of literature showing the different oncogenic and anti-oncogenic roles that lncRNAs play in CRC progression, such as lncRNA-cCSC1, which promotes proliferation [46]; TINCR, which regulates apoptosis [47]; and lnc-RI, which mediates radio-resistance [48]. lncRNA regulation occurs throughout tumor development, during the early stages such as SNGH11 [49]; in advanced stages, for example, MFI2-AS1 [50]; and even in metastatic tumors with both pro- and anti-tumoral effects, such as FTX and NBR2, respectively [51,52]. Thus, our results highlight the importance of researching lncRNAs in CRC because of their potential role as gene master regulators.

We found SCNAs with varying frequencies in all chromosomes with amplification (chr7, 8, 13, 20) and deletion (chr14, 15, 17, 18) peaks, both with several cases above 50%. Previous reports of SCNAs occurring frequently and in genomic locations associated with genes related to tumor progression in CRC corroborate our findings [53]; in turn, these results highlight the importance and occurrence of chromosome instability (CIN) in CRC and show the relevance of validating the presence and effect of alterations in the non-coding transcriptome.

Several of the 78 DElncRNAs we found have documented roles in other cancers, such as EXOC3-AS1, which has been reported to be upregulated in lung cancer patients [54,55]; INHBA-AS1, proposed to be part of a diagnostic marker in gastric cancer [56] and proven to promote CRC via sponging miR-422a [57]; and CRYM-AS1, associated with the overall survival of thyroid cancer patients [58] and identified as a potential tumor suppressor in gastric cancer, due to its negative regulation of CRYM [59]. The fact that, to our understanding, there are no previous reports of said lncRNAs in CRC underscores the necessity of continuing research into the effect of SCNAs on non-coding genomic regions and lncRNAs, especially since other groups have analyzed TCGA datasets in search of DElncRNAs and found very different results. For instance, Liu and collaborators [60] found a completely different set comprising 22 upregulated and eight downregulated lncRNAs. Yuan and colleagues [61] found a correlation between tumor progression and eight upregulated lncRNAs, none of which were in our list.

When we constructed the term–gene graph for the GO analyses, we found several enriched transcription factors (TFs): SOX4, MYCN, and MYBL2, among others. Increased SOX4 expression is associated with CRC in the tissue samples of patients, inhibiting both proliferation and invasive potential in CRC cell lines [62]. SOX4 directly interacts with HDAC1, which, in turn, maintains stemness in HCT-116 cells via the Wnt and Notch pathways [63]; downregulated SOX4 inhibits CRC progression [64] and metastasis [65]. MYCN has been amply described in central nervous system malignancies like neuroblastoma [66] and ependymoma [67]. In CRC, it has been reported that the MYCN pathway is upregulated [68], and even though its role has not been established, it might resemble that in other malignancies. MYBL2 has been proposed as a prognostic biomarker in CRC due to its role in proliferation, cell cycle progression, apoptosis, and overall poorer disease-free survival in patients [69]. MYBL2 is a key oncogenic TF for cell proliferation, DNA synthesis, and cell cycle progression. It binds to the promoter of a ribonucleotide reductase subunit, actively promoting its transcription through the S-phase of the cell cycle in CRC cell lines [70].

The network reported herein modulated the cell cycle via E2F1 and E2F5. E2F1 has been reported as oncogenic, because it promotes the cell proliferation of CRC cell lines [71], and upregulated in CRC patients, acting as an oncogene regulated by miR-326 [72]. Likewise, E2F5 is regulated by a competing endogenous lncRNA, SNHG6, via sponging miR-181a-5p [73] or downregulated by miR-34a [74], in both cases suppressing CRC progression. Upregulated E2F5 is associated with the positive regulation of TGF-β signaling in prostate cancer [75], which might also occur in CRC. On the other hand, there are also plenty of reports of TFs acting as tumor suppressors in CRC; among them is FOXO3, whose downregulation correlates with a better prognosis in CRC patients [76,77]. Another example is RUNX3—this TF is a target of the TGF-β pathway and is downregulated in CRC patient samples [78]. It has been reported that RUNX3 can induce apoptosis [79] and inhibit metastasis and angiogenesis [80]. The role of deregulated TFs in CRC is very well described elsewhere [81].

Overall, our results showed that one of the most important roles of our ceRNet was mediating transcriptional regulation through increased TF activity; nonetheless, this is not the only potential mechanism through which lncRNAs regulate transcriptional activity.

The GO analysis with the downregulated mRNAs in the described ceRNet (Figure 6B) showed that the most enriched process was the delayed rectifier potassium channel (cKv) activity. CKvs have been associated with tumoral processes. In stomach cancer cell lines, cKvs are upregulated and involved in proliferation via regulating $Ca^{2+}$ entry [82]. CKvs are also involved with the migratory and invasive capabilities of tumoral cells [83,84]. There are two particular cKvs with tumor-suppressor capabilities, Kv1.1 and Kv1.3; their upregulation is associated with tumor sensitization to different cytotoxins and lower cell survival [85]. Another relevant cKv, KCNA1, favors oncogene-induced senescence and lower aggressiveness in breast cancer cells [86]. The role of KCNA1 as a potential early-stage CRC biomarker suggested by the TCGA data has been validated in a 200-patient independent cohort [87].

HOTAIR is perhaps the most studied and reported lncRNA; from our DElncRNAs, only HOTAIR and LINC00461 have been reported in a similar ceRNA network [88]. In concordance with previous reports, high HOTAIR expression levels showed a poorer outcome. HOTAIR has been described as an onco-lncRNA because of its differential expression in several types of malignancies and its role in proliferation, migration, resistance to apoptosis, and pharmacological resistance [89–92]. HOTAIR can exert regulatory functions through multiple interactions with PRC2 [93] and through lncRNA–miRNA–mRNA axes, such as mir-326, mir-197, and mir-203a-3p [94–96]. Conversely, we found that low RRN3P3 expression was associated with lower overall survival. RRN3P3 has been reported as an oncogenic pseudogene in breast cancer due to its correlation with lower overall survival in high-risk patients in combination with other pseudogenes, especially in patients with the basal-like subtype [97]. This discrepancy might arise due to the high degree of tissue-specificity of lncRNA expression, which leads to closer expression patterns in related tissues (e.g., stomach, colon, and small intestine) than in unrelated tissues [98]. Thus, our results highlight the importance of HOTAIR as an onco-lncRNA that can act through diverse mechanisms, not only through endogenous competition for miRNA expression.

Proteoglycans in cancer have been detected as both oncogenic and anti-oncogenic. For example, Syndecan1 (a membrane proteoglycan, SDC1) loss is associated with CRC development and clinical stage [99]. Previous work showed that SDC1 overexpression regulates cell proliferation in CRC cell lines via suppressing CyclinD1. Furthermore, SDC1 inhibits cell migration, hindering MMP-9 and even blocking the JAK1/STAT and Ras/Raf/Mek/Erk pathways [100]. A report showed that low SDC1 expression was associated with metastatic potential, tumor recurrence, and shortened overall survival in CRC patients [101]. Since proteoglycans possess high structural heterogeneity, research focused on them is usually challenging [102].

Transcriptional alterations during tumor progression via oncogenic addiction are essential to maintain mutations arising in the early stages through to the advanced stages

due to their central roles in transcriptional programs that regulate key cell functions [103]. These results point out the relevance of lncRNAs as master regulators of gene expression, as here we found bioinformatic evidence of their involvement in major transcriptional regulation pathways.

The enrichment in the cellular response to estradiol pathway was explained by the fact that, in CRC, estradiol-17β (E2) interacts with its receptor and acts as a TF modulating the expression of multiples genes bearing estradiol response elements [104]. Estradiol receptors (ERs) can be oncogenic or tumor-suppressive depending on the receptor expressed. If E2 interacts with ERα, it has tumorigenic potential via the activation of the Wnt/β-catenin pathway [105]; conversely, interaction with ERβ has a tumor-suppressing role [106]. Whilst these results suggest the importance of E2 in CCR, it is still necessary to establish which receptor mediates this interaction and, therefore, the effect it elicits in CRC.

Another important molecular player affected by our ceRNA network (Figure 6D) was ERbB4, which has been reported to be downregulated in CRC tissue samples such as ours. Its loss might correlate with CRC progression [107]; nevertheless, ErbB4 has been reported as overexpressed in differentiated CRC cell lines and correlates with increased survival and growth [108].

When repeating the analysis including the full DEmRNA set, we found enriched 3′-UTR-mediated mRNA destabilization. mRNA destabilization occurs when an miRNA binds to the 5′ mRNA region [109]. It is estimated that 3′ UTR destabilization represents the majority of miRNA-mediated repression (60–90%), independently of the translational state; this has been observed in a variety of cell culture systems [110]. There are multiple reports of a tumorigenic effect due to miRNA-mediated 3′-UTR destabilization, with the main focus on constructing ceRNets [60,111–113]. Furthermore, we find it relevant to emphasize that this is the mechanism underlying the ceRNA network hypothesis, and so these results contribute to the main narrative of this work, i.e., that lncRNAs act as central players in transcriptional regulation.

This work opens several avenues for further research. First, we foresee the deepening relevance of proteoglycans in cancer thanks to their involvement in many biological processes. Second, the reports of lncRNAs with both oncogenic and tumor-suppressor roles depending on the context [114–116] imply that RRN3P3 can act as an oncogene in breast cancer and as a tumor suppressor in CCR; this would require experimental validation to better understand the role of this lncRNA in determining tumor phenotype. Moreover, there are reports of lncRNAs with double [117,118] and complex transcription sites, i.e., inside the loci of other ncRNAs, which require more experimental validation to shed light on their role in general and in CRC.

Overall, our results showed the relevance of SCNA-associated lncRNAs for colon cancer phenotype and several novel potentially oncogenic and tumor-suppressing lncRNAs. The completely bioinformatic nature of this approach represents its main limitation; our results should be validated in the future with experimental approaches such as microarray analyses for detecting both the SCNAs and SCNAs-associated lncRNAs, qPCR for lncRNA expression, RNA-pulldown for lncRNA-miRNA interactions, and CLIP assays for miRNA–mRNA interactions. These experiments could be complemented with functional assays (evaluating proliferation, migration, apoptosis, etc.) to shed light on the effect of the expression exerted by the SCNA-associated lncRNAs in the CRC phenotype.

**Supplementary Materials:** The following supporting information can be downloaded at: https://www.mdpi.com/article/10.3390/cimb45120597/s1, Figure S1. Profile of deregulated miRNAs in CRC. Figure S2. Profile of mRNAs deregulated in CRC. Figure S3. CCDC144NL is associated with five-year outcome in patients with CRC. Figure S4. LINC00479 is associated with 5-year outcome in CRC patients. Figure S5. TSPY26P is associated with 5-year outcome in CRC patients. Figure S6. C5orf64 is associated with 11-year prognosis in CRC patients. Table S1. List of lncRNAs encoded in altered regions in CRC patients. Table S2. List of mRNAs and miRNAs regulated by lncRNAs in the competing endogenous RNA network (ceRNet).

**Author Contributions:** C.P.-P. and H.H.-O. conceived the study; V.G.-C., E.L.-U., M.R.-D. and A.D.M.-G. obtained data to perform the analysis; H.H.-O. performed said analysis; D.C.d.L., C.L.-C., O.M.-C., N.J.J.-H., E.P.-Y. and R.R.-P. critically reviewed the manuscript; C.P.-P. and H.H.-O. wrote the manuscript. All authors have read and agreed to the published version of the manuscript.

**Funding:** H.H.-O. was the recipient of a CONACyT scholarship (1096935); V.G.-C. was funded by EDOMEX-FICDTEM-2022-01 (COMECYT-443.01-E34-19).

**Institutional Review Board Statement:** Not applicable.

**Informed Consent Statement:** Not applicable.

**Data Availability Statement:** All data are available from the TCGA website (https://www.cancer.gov/ccg/research/genome-sequencing/tcga (accessed on 15 March 2022); our results are provided in the Supplementary Materials.

**Conflicts of Interest:** The authors declare no conflict of interest.

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
