# Peer review of "Somatic Copy Number Alterations in Colorectal Cancer Lead to a Differentially Expressed ceRNA Network (ceRNet)"

_cimb, doi:10.3390/cimb45120597_

Round 1
Reviewer 1 Report
Comments and Suggestions for Authors
In the manuscript ID cimb-2701733, the authors report a study of the lncRNA whose expression is altered together with somatic copy number alterations in colorectal cancer (CRC) and how they are organized in a differentially expressed ceRNA network (ceRNET). Overall, the idea at the base of the research is good and the results are sufficiently interesting for the readers, although most of the discoveries reported are already part of the knowledge about CRC. However, there are a number of major and minor flaws, especially in the reported bibliography, that need to be fixed before a decision on publication.
Major concerns, including bibliographic issues.
1. The bioinformatics workflow (lines 94-96) will greatly improve by adding a scheme illustrating it, besides the M&M description.
2. Supplemental Tables 1 and 2 are not present in the available material, so can not be evaluated.
3. In lines 106-110, did the authors check for bifunctional RNAs? See for example these works: PMID: 37777390; PMC: 4683094; PMC: 4870183. In addition, did they check for ncRNA harboring other ncRNA? See for example PMID: 31014355 regarding CRC. These works (and other similar) should be added to the introduction and/or discussion, and the authors should explain if they searched for them or not, or if they excluded them and why.
4. Lines 126-127 and other locations (e.g., discussion) – did the authors check if the amount of lncRNA alterations is linearly or exponentially linked to deletion/amplification? In other words, are these alterations merely a byproduct of having more (or less) copies than normal of these gene? Indeed, one would expect that if I have (for example) six copies of a gene instead of the expected two, I will have 3x the expression of that gene compared to controls. This does not necessarily justify a cause/effect relation between lnc expression and CRC development, especially if in the same region maps an oncogene/oncosuppressor. Moreover, the authors do not report if the amplified/deleted regions indeed harbor “regular” genes connected with CRC formation/growth. These issues are not illustrated in the methods and results, nor discussed.
5. Line 151 - some lncRNA act directly on mRNA (e.g., antisense). Other alter gene expression by modifying chromatin (see for example PMC: 9146199). Did the authors check them? If yes, it is not clear. If not, why?
6. In line 304, the authors write “We found several lncRNAs with documented roles in other cancers”. It is not clear to this reviewer why the authors do not compare their results with similar studies performed on CRC, studies that are not cited in the references list. Some examples (but there are many others): PMID: 30503344; PMID: 31490563; PMID: 36272991; PMID: 30734239; PMID: 32377269; PMID: 37161577. The authors should cite these (and similar) works, compare their results with these works, highlight similarities and differences, and discuss them.
7. Line 336-337: the authors should cite and discuss this work: PMID: 32538588.
8. Line 370: the authors should cite and discuss this work: PMID: 33078631.
9. Overall, the discussion is too long and mostly a repetition of the results. This should be fixed. An analysis of how their results will be beneficial for CRC characterization and possibly the use of these data for diagnosis or for the identification of possible therapeutic targets would greatly improve its quality.
10. A three-lines “conclusions” section is meaningless. Either improve or delete.
Minor concerns.
1. In line 109, I suppose “y” stands for “and”. Please check.
2. Line 175 – please be consistent with lists – all numbers or all letters (“11”). Same in line 266.
3. Figure 1 (and similar, supplemental figures) are barely readable. I suggest to move scale colors below and increase size.
4. Figure 2 – to improve readability, I suggest to add the cited 10% threshold (a horizontal line will suffice) and to shorten the Y axis to 60-75%; there is no need to use an axis set to 100% if none of the data is above 50-60%.
5. Figure 3 is unreadable even when zooming in. Either provide a bigger image or add a supplemental table listing the ceRNET members.
6. Figure 4 is barely readable even after zooming in. In addition, please check consistency of panels identification, presently a mix of letters (figure) and numbers (figure legend).
7. Figure 5, barely readable as well.
8. Line 259: what do you mean by “Fig.S3:6”? Please check.
Author Response
Responses to Reviewer 1
In the manuscript ID cimb-2701733, the authors report a study of the lncRNA whose expression is altered together with somatic copy number alterations in colorectal cancer (CRC) and how they are organized in a differentially expressed ceRNA network (ceRNET). Overall, the idea at the base of the research is good and the results are sufficiently interesting for the readers, although most of the discoveries reported are already part of the knowledge about CRC. However, there are a number of major and minor flaws, especially in the reported bibliography, that need to be fixed before a decision on publication.
Major concerns, including bibliographic issues.
- The bioinformatics workflow (lines 94-96) will greatly improve by adding a scheme illustrating it, besides the M&M description.
Thank you for the positive recommendation, a workflow diagram has been added to the manuscript as Figure 1
- Supplemental Tables 1 and 2 are not present in the available material, so can not be evaluated.
Dear reviewer, thanks for your kind comment, the supplementary tables were certainly missed, we have included in the revised versión.
- In lines 106-110, did the authors check for bifunctional RNAs? See for example these works: PMID: 37777390; PMC: 4683094; PMC: 4870183. In addition, did they check for ncRNA harboring other ncRNA? See for example PMID: 31014355 regarding CRC. These works (and other similar) should be added to the introduction and/or discussion, and the authors should explain if they searched for them or not, or if they excluded them and why.
Although it is an interesting approach, we excluded bifunctional RNAs as we decided to focus only on transcriptional regulation. Searching for small peptide coding potential is an independent and resource-intensive process (e.g., PMID: 26364619, PMID: 29643274) that falls outside the scope of our hypothesis.
- Lines 126-127 and other locations (e.g., discussion) – did the authors check if the amount of lncRNA alterations is linearly or exponentially linked to deletion/amplification? In other words, are these alterations merely a byproduct of having more (or less) copies than normal of these gene? Indeed, one would expect that if I have (for example) six copies of a gene instead of the expected two, I will have 3x the expression of that gene compared to controls. This does not necessarily justify a cause/effect relation between lnc expression and CRC development, especially if in the same region maps an oncogene/oncosuppressor. Moreover, the authors do not report if the amplified/deleted regions indeed harbor “regular” genes connected with CRC formation/growth. These issues are not illustrated in the methods and results, nor discussed.
This is an interesting question that would certainly validate our results. Nevertheless, it is not possible to analyze correlations between differential expression and the nature of the genomic alteration with the current datasets, as the genomic and transcriptomic data were sourced from different patient groups.
- Line 151 - some lncRNA act directly on mRNA (e.g., antisense). Other alter gene expression by modifying chromatin (see for example PMC: 9146199). Did the authors check them? If yes, it is not clear. If not, why?
LncRNAs are a vast group of transcripts with a broad spectrum of functions –including chromatin modifications– each of which requires independent analysis workflows. To be as thorough as possible in our analysis, we focused only on miRNA-mediated transcriptional regulation, that was our goal during the course of this project.
- In line 304, the authors write “We found several lncRNAs with documented roles in other cancers”. It is not clear to this reviewer why the authors do not compare their results with similar studies performed on CRC, studies that are not cited in the references list. Some examples (but there are many others): PMID: 30503344; PMID: 31490563; PMID: 36272991; PMID: 30734239; PMID: 32377269; PMID: 37161577. The authors should cite these (and similar) works, compare their results with these works, highlight similarities and differences, and discuss them.
References to the mentioned studies have been incorporated into the discussion. Still, our work is, to the best of our knowledge, the first to survey differentially expressed lncRNAs associated with SCNAs, which makes it fundamentally different from those brought up by the reviewer.
- Line 336-337: the authors should cite and discuss this work: PMID: 32538588.
Dear reviewer, unfortunately we did not have access to the recommended article; however, we reviewed again the scientific literature and included a large part of the most relevant articles in the area.
- Line 370: the authors should cite and discuss this work: PMID: 33078631.
The paper by Uhan and colleagues has been included in the discussion.
- Overall, the discussion is too long and mostly a repetition of the results. This should be fixed. An analysis of how their results will be beneficial for CRC characterization and possibly the use of these data for diagnosis or for the identification of possible therapeutic targets would greatly improve its quality.
Thank you for your insight. The discussion has been shortened, mainly by removing references to the results acquisition process.
- A three-lines “conclusions” section is meaningless. Either improve or delete.
The three-line conclusion has been removed.
Minor concerns.
- In line 109, I suppose “y” stands for “and”. Please check.
Thank you for pointing this out. The text has been corrected.
- Line 175 – please be consistent with lists – all numbers or all letters (“11”). Same in line 266.
Unless the editors disagree, it is customary to spell out digits and to use numerals for larger numbers in a continuous text. The mistakes in line 266 and elsewhere have been corrected; thank you for pointing them out.
- Figure 1 (and similar, supplemental figures) are barely readable. I suggest to move scale colors below and increase size.
- Figure 2 – to improve readability, I suggest to add the cited 10% threshold (a horizontal line will suffice) and to shorten the Y axis to 60-75%; there is no need to use an axis set to 100% if none of the data is above 50-60%.
- Figure 3 is unreadable even when zooming in. Either provide a bigger image or add a supplemental table listing the ceRNET members.
- Figure 4 is barely readable even after zooming in. In addition, please check consistency of panels identification, presently a mix of letters (figure) and numbers (figure legend).
- Figure 5, barely readable as well.
Thank you for your comments regarding the readability of the figures. They have been modified accordingly, both in the stand-alone and embedded versions.
- Line 259: what do you mean by “Fig.S3:6”? Please check.
This was meant to be Fig.S3-6. The text has been corrected.
Reviewer 2 Report
Comments and Suggestions for Authors
Colorectal Cancer is recognized as the second deadliest malignancy globally. Approximately 75% of patients with Colorectal Cancer exhibit elevated levels of Chromosome Instability, leading to the accumulation of Somatic Copy Number Alterations (SCNAs). These alterations are known to be associated with the amplification of oncogenes and the deletion of tumor suppressor genes, contributing to the tumoral phenotype in various malignancies. Despite the well-established understanding of this relationship, there remains a significant gap in our knowledge regarding the impact of these alterations on long non-coding RNAs (lncRNAs) and, subsequently, their influence on the tumor phenotype. The present study aims to assess the role of differentially expressed lncRNAs located in regions with copy number alterations in patient samples with colorectal cancer.
The research team accessed RNA-seq files from the Colorectal Adenocarcinoma Project within the TCGA repository, which encompassed 285 sequenced tumor tissues and 41 non-tumor tissues. They performed an analysis of differential expression and aligned the identified genes with genome sequencing data, specifically focusing on regions with copy number alterations.
The study revealed a total of 78 differentially expressed lncRNAs, along with 410 miRNAs and 5028 mRNAs, meeting the criteria of Log2 Fold Change (LFC) greater than 1 or less than -1 and an adjusted p-value (padj) of less than 0.05. Subsequently, the research team constructed a competitive endogenous RNA (ceRNA) network, predicting significant interactions between lncRNAs, miRNAs, and mRNAs. This network comprised 30 lncRNAs, 19 miRNAs, and 77 mRNAs. Functional enrichment analysis, using KEGG and GO, revealed the involvement of these molecular components in various oncogenic and anti-oncogenic processes.
The findings of this study demonstrate that lncRNAs coded in regions affected by SCNAs play a pivotal role in forming a complex gene regulatory network in Colorectal Cancer.
The manuscript is interesting and worthy of publication in CIMB. However, it is important to note that the authors are recommend confirming these biostatistical findings through additional research conducted on their own patient cohort.
Author Response
The manuscript is interesting and worthy of publication in CIMB. However, it is important to note that the authors recommend confirming these biostatistical findings through additional research conducted on their own patient cohort.
Dear reviewer, we appreciate your comment, but we would like to explain in detail why we did not perform an experimental validation on patient samples. The results of this work showed some lncRNAs clearly involved in colorectal carcinogenesis; such as lncRNA-cCSC1, lnc-RI, SNGH11, MFI2-AS1, HOTAIR among others. The fact that we found previously characterized lncRNAs in colorectal carcinogenesis is evidence that the results of this work are consistent with the scientific literature. In addition, we also found lncRNAs that had not been characterized in this tumor, such as EXOC3-AS1, INHBA-AS1, CRYM-AS1 among others that could have an important role in the development of CRC. We are interested in characterizing the functional role and clinical utility of these previously uncharacterized lncRNAs. Future work will validate their role in the development of colorectal carcinogenesis.
Round 2
Reviewer 1 Report
Comments and Suggestions for Authors
Check lines 435-436 for language, the sentence is not clear.